# Cellulose Nanofibers from *Schinus molle*: Preparation and Characterization

**DOI:** 10.3390/molecules27196738

**Published:** 2022-10-09

**Authors:** Abir Razzak, Ramzi Khiari, Younes Moussaoui, Mohamed Naceur Belgacem

**Affiliations:** 1Laboratory for the Application of Materials to the Environment, Water, and Energy (LR21ES15), Faculty of Sciences of Gafsa, University of Gafsa, Gafsa 2112, Tunisia; 2Facultyof Sciences of Gafsa, University of Gafsa, Gafsa 2112, Tunisia; 3Laboratory of Environmental Chemistry and Clean Process (LCE2P-LR21ES04), Faculty of Sciences of Monastir, University of Monastir, Monastir 5019, Tunisia; 4Department of Textile, Higher Institute of Technological Studies (ISET) of Ksar-Hellal, Ksar-Hellal 5070, Tunisia; 5University of Grenoble Alpes, CNRS, Grenoble INP, 38000 Grenoble, France; 6Organic Chemistry Laboratory (LR17ES08), Faculty of Sciences of Sfax, University of Sfax, Sfax 3029, Tunisia

**Keywords:** *Schinus molle*, cellulose nanofibers, enzyme fiber care R, mechanical proprieties

## Abstract

*Schinus molle* (SM) was investigated as a primary source of cellulose with the aim of discovering resources to generate cellulose nanofibers (CNF). The SM was put through a soda pulping process to purify the cellulose, and then, the fiber was treated with an enzymatic treatment. Then, a twin-screw extruder and/or masuko were utilized to help with fiber delamination during the nanofibrillation process. After the enzymatic treatment, the twin-screw extruder and masuko treatment give a yield of 49.6 and 50.2%, respectively. The optical and atomic force microscopy, morfi, and polymerization degrees of prepared cellulosic materials were established. The pulp fibers, collected following each treatment stage, demonstrated that fiber characteristics such as length and crystallinity varied according to the used treatment (mechanical or enzymatic treatment). Obviously, the enzymic treatment resulted in shorter fibers and an increased degree of polymerization. However, the CNF obtained after enzymatic and extrusion treatment was achieved, and it gave 19 nm as the arithmetic width and a Young’s modulus of 8.63 GPa.

## 1. Introduction

Worldwide ecology is pushing researchers and industrialists to exploit new rapidly renewable, abundant and biodegradable sources. Lignocellulosic biomass is regarded as a promising material for replacing synthetic polymers and petrochemical-based goods due to its recyclability, availability, and affordability [1,2,3,4,5]. The manufacture of lignocellulosic material as a substitute for wood sources is among the most intriguing ways tovalue cellulose fibers [3,6,7,8]. Cellulose is one of the most important natural polymers on Earth, and it can play an important part in worldwide ecology, since it is produced in quantities that surpass 10^12^ tons per year [4,7,8,9,10,11,12,13,14,15,16,17,18].

Nowadays, the known product nanocellulose is considered an excellent biopolymer that has a large field of applications, such as biomedicine [6,19,20], food packaging [21,22], electrochemical performance [23], and as a giant reinforcement matrix [24,25]. Two kinds of nanocellulose, cellulose nanocrystals (CNC) and cellulose nanofibers (CNF), have been extensively researched in terms of isolation, characterization, and application. In contrast to synthetic polymers, nanocellulose is a fully renewable, durable, and lightweight material; inexpensive to produce; and safer to use. The preparation of CNF was attained utilizing a slew of chemical pretreatments, notably TEMPO-mediated oxidation, sulfonation, phosphorylation, carboxymethylation, etc. [18,26,27,28]. However, the high upfront expenditures of these pretreatments and the challenges associated with recycling these compounds constrain their further usage. The energy-and chemical-safebiological pretreatment found in 2007 now has great promise inindustrial applications due to its environmentally benign conditions and antigenicity of the enzymatic response [10,17,29,30,31]. Since then, enzymes including xylanase [32] and lytic polysaccharide monooxygenases [33] have been suggested to increase the pretreatment’s effectiveness. Numerous researchers [11,29,31,34,35] have concluded that cellulose pretreatment with cellulolytic enzymes cleaves glucosidic linkages and therefore favors and facilitates the fibrillation of cellulose fibers. The production of CNF involves mechanical treatment for delaminating the fibers [36,37,38]. The currently utilized mechanical equipment can involve high-pressure homogenizers, microfluidizers, specialized grinders, and twin-screw extruders [11,39,40]. Along with low density and thermal expansion, CNF have advantages in the development of nanocomposite materials that may be employed in a wide range of applications. CNF can find applications in water purification and drug delivery [41,42,43,44,45,46,47]. Diverse plant sources can be used for the production of CNF.

The goal of this work is to investigate the viability of generating CNF by treating bleached *Schinus molle* pulp with enzymes followed by mechanical treatment. The supramolecular structure, morphology, and physicochemical properties of the obtained CNF were studied.

## 2. Materials and Methods

### 2.1. Chemicals Products and Raw Material

All chemicals were employed as given by the manufacturers: acetic acid (99.7%, Sigma-Aldrich, Grenoble, France), deionized water, Fiber Care R enzymes (Novozymes, Denmark, 4770 ECUg^−1^, provided by the partner company Arjowiggins, France), sodium hydroxide (Sigma-Aldrich, Grenoble, France), and sodium hypochlorite (Sigma Aldrich, Grenoble, France).

*Schinus molle* was used as raw material and obtained from the Gafsa region (south of Tunisia). The chemical composition of *Schinus molle* was determined by applying the standard methods. *Schinus molle* has a relatively high content of cellulose, 53.2%, followed by lignin (21.4%) and hemicelluloses (21%). The amount of extractive solubility in hot water, in 1% NaOH, and in ethanol–toluene was 38.2, 27.0, and 12.3%, respectively. The ash content was about 4% [48]. The quite large content of cellulose allowed for envisaging the development of such a biomass as a source of cellulose for the production of CNF.

### 2.2. Delignification-Bleaching Step

The cellulose was isolated from the lignocellulosic biomass, namely “*Schinus molle*” (SM). The lignocellulosic material was collected, washed, and dried at 105 ± 5 °C to a constant weight. Then, the dried material was chopped into uniform pieces about 5 cm in size. Then, for each test, 60 g of raw material was delignified with 10 wt% NaOH at 160 °C for 2 h [48,49]. The resulting fibers were washed thoroughly with distilled water until they reached a neutral pH and bleached with a solution of sodium chlorite at 80 ± 5 °C for 3 h. The isolated cellulose was rinsed with distilled water until it reached a neutral pH. Finally, the obtained pulp (35.3 g) was then stored at 4 °C and used to prepare the CNF.

### 2.3. Enzymatic Pretreatment

The enzymatic pretreatment of bleached cellulose was carried out with the cellulolytic enzyme “Fiber Care R” at a dose of 300 ECU per 1 g of cellulose. The 2 wt% cellulose dispersion in acetate buffer solution (pH = 5) was treated with the enzyme solution while stirring at a speed of 200 rpm at 50 °C for 2 h. After that, the dispersion was heated to 80 °C for 15 min and then cooled to 25 °C. Finally, the treated cellulose was washed thoroughly with distilled water until a neutral pH and isolated using a 1-μm nylon mesh filter and stored at 4 °C for the further characterization and preparation of CNF.

### 2.4. Mechanical Disintegration

The fibrillation process was established with dual-screw extruder and by a masuko device. The Thermo Scientific HAAKE Rheomex OS PTW 16 + HAAKE PolyLab OS Rheo Drive 7 twin-screw extruder with a 16-mm diameter and 640-mm length was employed. The selected configuration has been reported before by our teams [48,50,51,52]. A force feeder, the Thermo Scientific PolyLab OS 567-4002, was utilized to feed the extrudes. An average dry flow rate of 123 ± 37 g h^–1^ was achieved with a feed rate of 2% atmaximum speed and a screw speed of 400 rpm. To prevent heating in the kneading region, pulp clogging from friction, and a change in dough wetness, the temperature was regulated by a cooling system with a fixed point of 10 °C. The dough underwent a total of 7 passes.

Fibrillation was also realized using a grinder, model MKZA6-2, disc model MKG-C 80, Masuko Sangyo Co., Ltd., Saitama, Japan. The process was done as previously reported process by Rol et al. [52] and Bettaieb et al. [48]. The obtained CNF using a dual-screw extruder and masuko device were designated CNF-TSE and CNF-M, respectively.

### 2.5. Nanopapers Preparation

To gain nanopapers, 2 g (of the dry matter CNF) are scattered and diluted with deionized water to 0.5% (*w*/*w*) and mixed for 30 min under magnetic stirring. At that point, the suspension is filtered using a Rapid Köthen device with the former handsheet equipped with a nylon sieve of 1-μm mesh size under vacuum. The obtained nanopapers are dried between two nylon sieves under vacuum at 45 °C for 3 h. The nanopapers are stored in a conditioned room at 23 °C and 50% relative humidity for 48 h before any characterization. Water was chosen as the diluent in this work for several reasons; the nanopaper sheets were prepared in order to evaluate the mechanical proprieties of the prepared CNF, water is considered a green chemical that provides for the environment, and it has no side effects and evaporates quickly.

Figure 1 shows the scheme for the preparation of CNF.

### 2.6. Characterization

Several tools were employed to characterize the fibers, the treated fibers, and the obtained CNF, as well as all the prepared nanopapers. A light microscope Carl Zeiss Axio Imager M1m was used. An Axio Cam MRc5 digital camera was used to take pictures of a drop of cellulose dispersion (0.2% by weight) inserted between the glass slide and cover slip. The Morfi LB-01 fiber analyzer (Techpap, Grenoble, France) was used to determine the cellulose fibers size distribution and fine contents. In the image capture device, a pulp dispersion (3 mg L^−1^) was circulated continuously. The fine contentswereassessed in % in length and fines/g.

The morphological structure of CNF was studied using transmission electron microscopy (TEM) and atomic force microscope. For the TEM analysis, a drop of diluted dispersion (0.1 weight percent) after 24 h of sitting was dropped on a copper microscope grid that was dusted with a coating of amorphous carbon-containing holes. Once all the liquid phase was evaporated, the grid was observed at room temperature with a HITACHI H-7000 transmission electron microscope. The AFM analysis was performed using the tapping mode of an OTESPA cantilever. A CNF dispersion (10^−3^ weight percent) was spread out on a mica disk and allowed to air dry. The measurements of the nanofibril diameters from the height profiles of the AFM images were done using the Nanoscope III program. Two hundred heights were assessed for each sample, which were scanned in at least four separate locations. From the TEM and AFM micrographs, the dimensions of the corresponding CNF were evaluated by a digital image analysis using a minimum of 100 CNF.

The thermal properties were established by using the thermogravimetric analysis (TGA). The measurements were performed using a STA-6000 thermogravimetric analyzer (Perkin Elmer Instruments, Villebon-sur-Yvette, Grenoble, France). In fact, about 10 mg of samples were dynamically scanned between 25 and 700 °C under a nitrogen stream (50 mL min^−1^) with a 10 °C per min^−1^ heating rate.

The wide-angle XRD spectra were recorded thanks to a PANanalytical X’Pert PRO MPD diffractometer with an X’celerator detector. The crystallinity index (CI) was determined according to the Segal et al. method [53]:(1)$$\mathrm{CI}\left(\%\right)=\frac{\left(I_{002}–{\;I}_{\mathrm{am}}\right)}{I_{002}}\times 100$$
where I_002_ denotes the primary crystal peak’s diffraction intensity and I_am_ the minimum intensity.

Scherrer’s equation was used to estimate the cellulose crystallites’ average thickness from their X-ray diffraction patterns [54]:(2)$${\;D}_{\mathrm{hkl}}=\frac{K\lambda }{\beta_{1/2}\cos\theta }$$
where D_hkl_ is the crystallite dimension normal to the hkl family of lattice planes, K is the correction factor, which is usually 0.94, λ is the radiation wavelength, θ is the diffraction angle, and β_1/2_ is the peak width at half-maximum intensity. Perpendicular to the 0 0 2 planes, the crystal size was determined.

According to the standard ISO 5351:2010, the intrinsic viscosity (η_int_) was used to calculate the degree of polymerization (DP). The Mark–Houwink–Sakurada equation was involved to determine the average viscometrical degree of polymerization (DPv) according to the following equation:DP_v_^0.905^ = 0.75 η_int_(3)

In order to determine the nanometric fraction, 400 mL of water were used to dilute 0.4 g of CNF. A centrifuge was used to spin the resulting dispersion at 1000 rpm for 15 min. The nanoscale fraction was estimated according to the method of Naderi et al. [55]. Three tests were run, and the average value was then found. An AL 250 T-IT AQUALYTIC turbidimeter was utilized to quantify the turbidity using CNF dispersions at 0.1 weight percent. There were 10 measurements, and the average was given.

The quality index (QI*), which was developed by Desmaisons et al. [56], was also determined. It characterizes the cellulose nanofibers. The QI* was calculated using the following equation:QI* = 0.3x_1_ − 0.03x_2_ − 0.072x_3_^2^ + 2.54 x_3_ − 5.34 ln(x_4_) + 58.62(4)
where x_1_ = nanosized fraction (%), x_2_ = turbidity (NTU), x_3_ = Young’s modulus (GPa) of the CNF nanopaper, and x_4_ = macro size (µm^2^) measured on microscopy images.

### 2.7. Properties of the Manufactured Papers

A large number of tests were done in order to characterize the prepared nanopapers. The evaluation of the surface mass was established according to the standard ISO 536. The grammage corresponds to the ratio between the mass of a nanopaper and its surface. The weighing of the paper was carried out using a precision scale. The grammage (G, g m^−2^) was given by Equation (5):(5)$$G=\frac{m}{S}$$
where m is the weight of the paper sheet (g), and S is the surface area of the paper sheet expressed in m^2^.

In addition, the thickness of the nanopaper sheets was measured. The protocol to determine the thickness (µm) of the nanopaper was described by the standard ISO-534. The principle of this measurement is based on the displacement of a piston: a difference in height is measured between a reference position (plate of the apparatus) and a measurement position (paper deposited on the plate). The measurement can be distorted if the nanopaper contains impurities or if the handsheet is very compressible. Ten measurements were made for each sample.

The mechanical properties of the nanopapers were tested. The measurements were carried out in a conditioned room according to the NF Q03-004 standard. Each strip of 10 × 1.5 cm previously cut wasblocked between the clamps of the device and then a force was applied on the mobile clamp to stretch the strip until its rupture. The results provided allowed determining the force and the elongation of a rupture, the Young’s modulus, and the length of rupture.

## 3. Results and Discussion

Cellulose fibers with high purity and crystallinity were extracted from *Schinus molle* using an alkaline and bleach treatment. From these fibers, different kinds of nanofibrils were prepared. In fact, an enzymatic treatment after the alkaline–bleach treatment has been explored. As such, the enzymatic treatment of cellulosic fibers has been reported in the literature [11,31,34,57,58]. It can improve the production of nanofiber kinetics and processing efficiency. The enzymatic pretreatmentof cellulose can reduce the degree of polymerization and fiber breakdown levels [59]. The amorphous domains of cellulose are hydrolyzed preferentially by cellulases, whereas the crystalline domains remain intact [11,34]. β-1,4-glycosidic linkages are shattered during the enzymatic hydrolysis of cellulose. This, in turn, leads to the generation of oligosaccharides with varying chain lengths [60]. The enzyme catalyzes the hydrolysis of a glycosidic bond via the participation of carboxyl groups and can occur through lattice inversion or an asymmetric carbon configuration [61]. This enzyme’s much increased endoglucanase activity (4770ECU/g) may account for the enhanced performance of cellulase Fiber Care R. The enzymatic treatment of cellulose isolated from *Schinus molle* previously treated with an alkaline solution gave a yield of about 51.2%. In addition, the NaOH treatment prior to enzymatic hydrolysis had a favorable impact. Precisely, the alkaline treatment appeared to make cellulose more accessible to enzymes [11,58], which enhanced the hydrolysis kinetics and yield [11,34,57,58]. These treatments were established in order to show that an alkaline pretreatment followed by enzymatic hydrolysis is efficient for producing CNF from *Schinus molle*. In this study, the alkaline pretreatment was followed by bleaching and an enzymatic step, and by mechanical fibrillation processes, to obtain nanoscale fibers. The effects of different treatment processes and the obtained cellulosic materials were well-studied and characterized.

The alkaline treatment used for cellulose extraction from *Schinus molle* caused changes in the fiber morphology. In the raw material and delignified fibers micrograph (Figure 2a,b), each elementary fiber displays a compact structure with a smooth surface, due to the presence of nonfibrous compounds, and exhibits alignment in the fiberaxis direction. Following cellulose extraction (Figure 2c), the surface became rougher. Moreover, the separation of individual fibers was observed, attributable to the removal of hemicellulose and lignin, which could be favorable for the accessibility of cellulase enzymes in the controlled hydrolysis of cellulose [11,35].

### 3.1. Physicochemical Characterization of Enzymatic Fibersand CNF

Generally, microfibrous dispersion is a mixture of two elements thatare microfibers and nanofibers (which are indicated by the presence of fine elements). The MORFI is an analyzer of the morphology of different fibers presented in microfibrous dispersion; it allows analyzing the elements present at amicrometric size. Additionally, this type of analysis allows giving an idea on the level of each compound present in the suspension. The procedure adopted considers a gross fiber as an element of length greater than 200 μm and fine elements as elements of lengths less than 200 μm.

The enzymatic hydrolysis process alters the shape of the fiber in significant ways. Its impact on untreated pulp is already apparent, as shown by drop-in fine contents from 38% to 29.7% and a reduction in fiber lengths from 618 to 408 μm (Table 1). After the two mechanical treatments, milling and extruder, fine elements and gross fibers are generated. Gross fibers with a width of 19 µm were obtained (Table 1).

For fibers treated with 10 wt% NaOH, this impact is amplified, resulting in an average fiber length of 618 μm. These findings confirm that the structural modifications brought on by enzymatic hydrolysis have an effect atthe micrometric scale. Here, the significant DP reduction is associated with a drop in the fiber length and cell wall destruction that produces fine components. Unexpectedly, it was discovered that the twin-screw (TSE) and masuko grinder steps after that were more effective than the NaOH and enzymatic treatments [62]. After enzymatic hydrolysis, the masuko grinder, and TSE treatments, the samples showed fiber lengths of 316, 340 μm, and significant fine element contents of 95.6 and 89.2%, respectively (Table 1). This proves the presence of a significant level of nanofibrils in the microfibril suspension mixture. These are detected subsequently by AFM and TEM. One can see from the short strands that they were effectively weakened and couldbe readily cut by hand. Meanwhile, the production of such minute components is indicative of the formation of cellulose microfibrils and, most likely, nanofibrils. The findings showed that the use of NaOH 10% improves the efficiency of converting cellulose I into cellulose II for enzymatic hydrolysis and its subsequent deconstruction by mechanical processes. These results further demonstrate the extensive fiber destruction that occurs throughout the enzymatic process, with shorter pieces of fibers being formed as a result.

In comparison to kraft Eucalyptus wood and vine stem fibers, the resultant CNFhas a smaller fiber length of 408 μm [63]. It demonstrated a comparable range when compared to other agricultural leftovers and hard woods [64] in terms of fiber width (19 μm). The fine percentage was 89.2%, greater than the industrial pulps utilized in the paper industry and consistent with other agricultural leftovers [65]. The manufacturing of fiberboards with better mechanical qualities may be made possible by a high fine contents level. As a result of virgin fines’ strong binding ability and smaller sizes compared to fibers, there is more contact between the fibers [66].

The polymerization degree of cellulose fibers decreased after the enzymatic treatment [67], which is a significant consequence. The increased availability of water inside the cell wall promotes swelling and solvation of the nanofibrils, which, in turn, aids the exfoliation of the fiber wall. As was noted before, endoglucanase enzymes are distinguished by the quick attenuation of the polymerization degree they induce. Table 1 shows that, after the enzymatic and mechanical pretreatments, the degree of polymerization drops significantly from 618 to 275. For a given level of mechanical and/or enzymatic processing, the DP reduces [11,34]. Comparable to other authors [67,68,69,70] who have reported producing CNF using similar enzymatic and mechanical pretreatments, the DP of cellulose is reduced by between 72% and 81% for the CNF generated, with DP values between 275 and 465. The drop in DP might possibly be attributed to the impact of enzymes on the pulp, which destroy the amorphous regions of cellulose [31]. It is also worth noting that the degree of polymerization of the cellulose chains decreased throughout the fibrillation process in the homogenizer. As the number of passes through the homogenizer increases, shear forces cause a higher cut of the fibers, causing the polymeric chains to break and their lengths (length and breadth) to be reduced, permitting the creation of microfibrils and nanofibrils. Henriksson et al. [31] discovered that using a homogenizer reduced the DP value by 30–50%. Unsurprisingly, and as previously documented [71], the DP value drops when the extruder is used. We saw a 59% reduction in DP for the enzymatic pulp after seven cycles with the pretreated pulps. According to these findings, there seems to be a clear link between the degree of cellulose polymerization, its ease of fibrillation, and the sizes of the nanomaterials. The degree of polymerization, on the other hand, is connected to the length of the cellulose chains and so offers information on the amount of cleavage in the fiber direction. As the degree of polymerization decreases, the fiber length decreases with each run through the extruder and/or grinder. Furthermore, like other mechanical method used to generate CNF, twin-screw extrusion, results in a drop in DP [72]. However, the enzymatic processing of cellulose fibers is responsible for the majority of the DP reduction.

Additionally, Table 1 shows the yields of cellulose, enzymatic fibers, CNF–TSE, and CNF–M, which explains the strength of the hydrolysis process, as well as the solubilization and enzymatic degradation of the carbohydrates. After the enzymatic treatment, the yield shifts from 58.8 to 51.2%. The yields of NFC–TSE and NFC–M are comparable to those of enzymatic pulp, indicating that the raw material had no significant effect on the yield of mechanically obtained CNF, at least with these treatments.

### 3.2. Morphological Properties of CNF

The morphology of CNF was studied by different methods, such as optical microscopy, a fiber analyzer, and AFM imaging.

The AFM imaging of CNF yielded the extremely microfibrillated nanofibers with a large aspect ratio (length/diameter) seen in Figure 3c,d. The nanofibers are several micrometers long and 20–30-nm wide. Additionally, several shorter nanofibers with an average thickness of between 5 and 10 nm are seen in the micrographs. The uniform size distribution was observed for the CNF. Even though the fibers were entangled by the long lengths, they were well-dispersed atalmost elemental levels of the fibrils (Figure 3c,d). This indicates efficient fibrillation and separation of the CNF by the combinations between the enzymatic and mechanical treatments [32].

From images captured by the optical microscope (Figure 3a,b), as the number of runs through the homogenizer increases, the enzymatic pulp seems to become more homogenous and smaller in size. A significant increase in the quantity of fibers was seen after processing. The breakdown of the cell wall caused by the hydrolysis and homogenization processes, which results in the formation of smaller particles and a rise in their number, was the cause of this increase. A small nanofraction of the CNF materials was observed (Figure 3a,b). The CNF may be achieved after six passes through a twin-screw extruder, as revealed by Ho et al. [71], and more runs result in deterioration.

The TEM images (Figure 3e,f) revealed that the extrusion process has a significant impact on cellulose fiber fibrillation. In general, the fibrillation impact increases with the number of passes. The seven TSE treatments resulted in heavily intertwined fibers. Other disintegration procedures, such as high-shear homogenization [39,40] or grinding, provided the most homogeneous materials (with and without the pretreatment) after around ten passes [39,40].

### 3.3. X-ray Diffraction Analysis of CNF

The super-molecular structures of enzymatically hydrolyzed celluloses were studied using XRD. Diffraction patterns for the cellulosic materials before and after the biotransformation process are shown in Figure 4. The X-ray diffraction patterns were analyzed for each sample to see how the cellulose crystallinity changed over time. The XRD pattern was characteristic of the I_β_ crystalline structure of cellulose (2θ = 15.5°, 16.7°, 22.5°, and 34.7°). The crystallinity index was found to be 54.8%, 58.1%, 70.2%, and 74.1% for the bleached fibers, enzymatic fibers, CNF–TSE, and CNF–M, respectively. Obviously, the enzymes attacked the amorphous zone and were found to degrade much faster than in the crystalline zone [11,34,73]. This fact shows that the sample’s crystallinity index increased while still retaining the crystalline allomorph typical of cellulose I.

The average cross-sectional dimension of the elementary cellulose crystallites was determined from X-ray diffractograms by applying Scherrer’s expression (Equation (2)). Since Scherrer’s equation is restricted to samples of high crystallinity and without any broadening of the peaks, this estimated calculation was made only for CNF-TSE and CNF-M, and the values were 19 nm and 12 nm, respectively. The value seems to be very close to the obtained width, which was determined by optical observation (AFM and TEM).

### 3.4. Thermogravimetric Analysis of CNF

While nanofibers show promise in a variety of areas, thermal stability is essential for their widespread use. Therefore, thermal stability of the *Schinus molle*, enzyme-treated *Schinus molle*, and CNF-TSE samples were investigated (Figure 5).

There were basically three areas in which the samples underwent pyrolysis: area I < 100 °C, 100 °C ≤ area II ≤ 400 °C, and area III > 400 °C. The first area had about a 4% decrease in the CNF, bleached, and enzymatic fiber weight due to water loss. The CNF, bleached, and enzymatic fibers’ main mass loss zone was area II, as a consequence of extensive dehydration and depolymerization at this stage. The CNF lost roughly 48% of their mass after this stage. However, for the bleached and enzymatic fibers, respectively, losses of 78 and 80% were seen. The CNF graphitization (area III) was primarily responsible for further degrading the products of area II with a loss of about 46%, resulting in molecules with low molecular weights [34]. In this area, the bleached and enzymatic fibers lost about 15–16% of their masses, which corresponded to the degradation of the cellulose chains.

### 3.5. Evaluation of Physical Properties of CNF Made from Schinus molle

Table 2 lists some physical characteristics of the various samples. The turbidity was measured as the amount of light transmitted, proposed as a quick method to evaluate the dispersion of cellulose nanomaterials [74]. Thus, lower turbidity of the suspensions means more fibrillated CNF. Based on its well-individualized fibers and lack of aggregates, the initial enzymatic pulp has a comparatively low turbidity (250 NTU). The turbidity was increased to 706 and 494 NTU for the CNF obtained by the twin-screw extruder and masuko grinder, respectively. These outcomes were brought about by the substantial quantity of aggregates produced by extrusion, as seen using an optical microscope. The enzymatic CNF suspensions obtained from TSE presented a value of turbidity (706 NTU) more than the CNF obtained from the masuko grinder (494 NTU), and the lower sizes of the individual enzymatic CNF wereconfirmed, regardless of the number of passes. The reported data were higher than the literature (242 NTU) [56].

There were noticeable variations in the pulp characteristics as a result of bioextrusion. The comparatively low-tear CNF resistance of 54.34 mN after 10 enzyme runs demonstrated efficient fiber cutting and the production of fine components. Yet, the masuko grinder led to a final value of 49.014 mN for tear resistance. The nanofibers produced thanks to TSE after ten passes and to the grinder (masuko) presented a Young’s modulus of about 8.63 and 12.31 GPa, respectively. These values were less than those of the commercial ones and the CNF made using a grinder from bleached Kraft pulp from softwood (Loblolly pine) (16 GPa) [75]. However, the Young’s modulus for the CNF prepared by TSE was also preferable compared to the 6.3 GPa for the nanopapers made from rice straw [76]. While both extrusion and grinding may reduce the mechanical characteristics, extrusion often has a more negative impact. In addition, the low Young’s modulus (8.63 GPa) of the CNF from TSE demonstrated that the fibers were really weakened. After a certain number of passes, the materials’ characteristics were either not improved or they degraded.

With a reduction in turbidity, the findings in Table 2 suggest 69% and 52% nanosized fractions for the TSE and masuko extruded fibers, respectively. These portions were lower than for commercial CNF (73 ± 14%) [55]. This indicates that, despite the relatively lengthy processing period, only the enzymatic fiber portions were defibrillated into nanofibrils. The evolution of the quality index showed a clear difference in the properties of the CNF for bioextrusion and the masuko grinder. The use of the TSE led to a quality index of 63.9%, while the use of masuko led to a slight increase in the quality index to 65.5%. Therefore, these CNF will be categorized as lower-quality accessible CNF [76].

## 4. Conclusions

Enzymatic treatment aids the mechanical shearing process that breaks down cellulosic fiber pulp into nanofibers. The enzymatic pretreatment, followed by two mechanical treatments, were used in this study to successfully produce cellulose nanofiber suspensions from alkali-treated bleached *Schinus molle* pulp. This treatment, which is employed by enzymatic hydrolysis using Fiber Care R endoglucanase solution because it results in less depolymerization and better nanofibril dissociation, presents many advantages, such as being environmentally friendly, solvent- and chemical-free, etc. Additionally, it was demonstrated that the processing conditions that were used, which had an impact on the material properties, affected the morphology of the CNF dispersions that were produced. Nanofibrils are smaller as a result of more intense fibrillation. The yields of the obtained CNF were about 49.6 and 50.2%, respectively, for CNF-TSE and CNF-M. The nanofibers created after the enzymatic pretreatment with masuko or TSE had a high aspect ratio, according to the AFM and TEM images. These nanofibers exhibited high aspect ratios and are highly sought after for use as reinforcements in nanocomposite materials. Following the enzymatic–mechanical treatments, the estimated DP dropped from 618 to 275. Whatever the mechanical treatment, the cellulose nanofibers presented a diameter range between 5 and 10 nm. It can be concluded that the combination of enzymatic and extrusion led to obtaining CNF with less fiber aggregation, which improved their mechanical properties. These can be interesting and promising nanomaterials for different applications. An investigation of the CNF as reinforcement elements is under study, and the paper will be reported very soon.

## Figures and Tables

**Figure 1 molecules-27-06738-f001:**
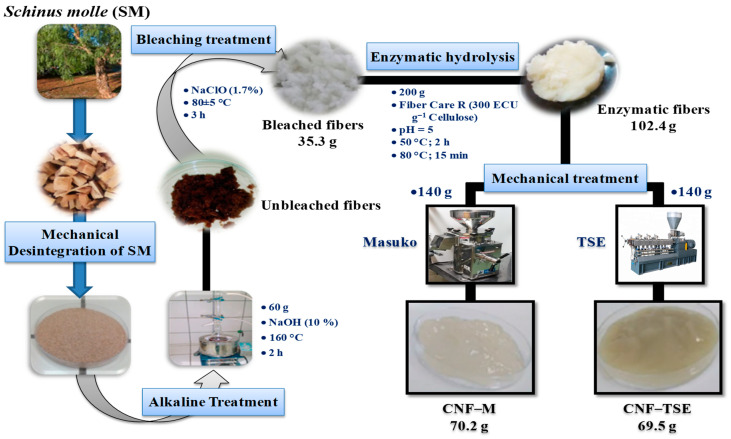
Schematic illustration of procedures used to produce CNF from *Schinus molle*.

**Figure 2 molecules-27-06738-f002:**
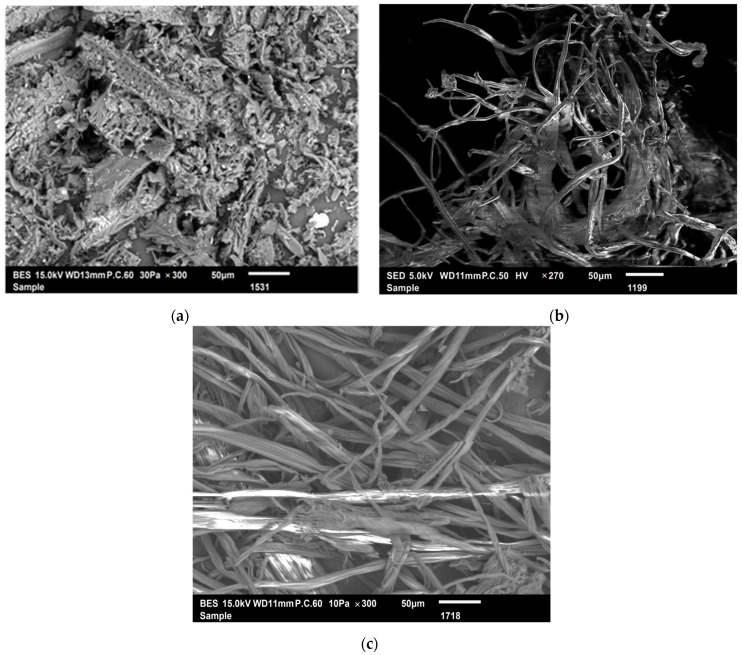
SEM observations of raw material (**a**), delignified fibers (**b**), and delignified, bleached fibers (**c**).

**Figure 3 molecules-27-06738-f003:**
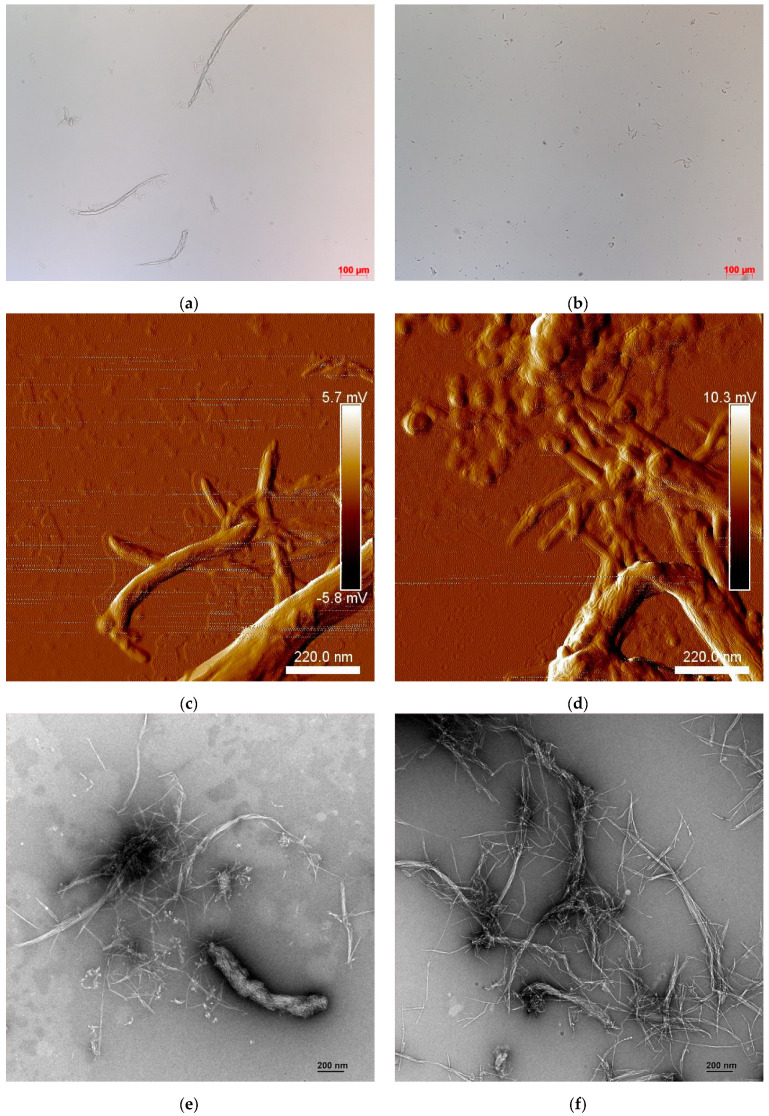
Optical observation of CNF–TSE (**a**) and CNF–M (**b**). AFM images of CNF–TSE (**c**) and CNF–M (**d**). TEM micrographs of CNF-TSE (**e**) and CNF–M (**f**).

**Figure 4 molecules-27-06738-f004:**
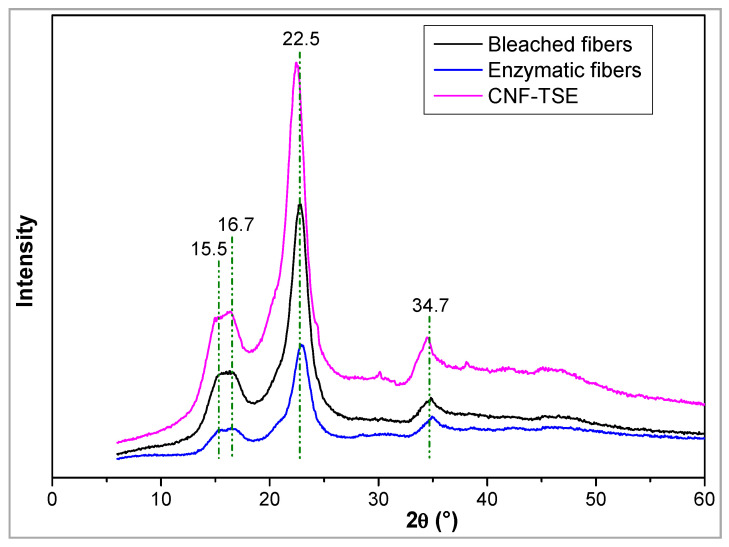
An example of XRD profiles of the bleached fibers, enzymatic fibers, and CNF–TSE.

**Figure 5 molecules-27-06738-f005:**
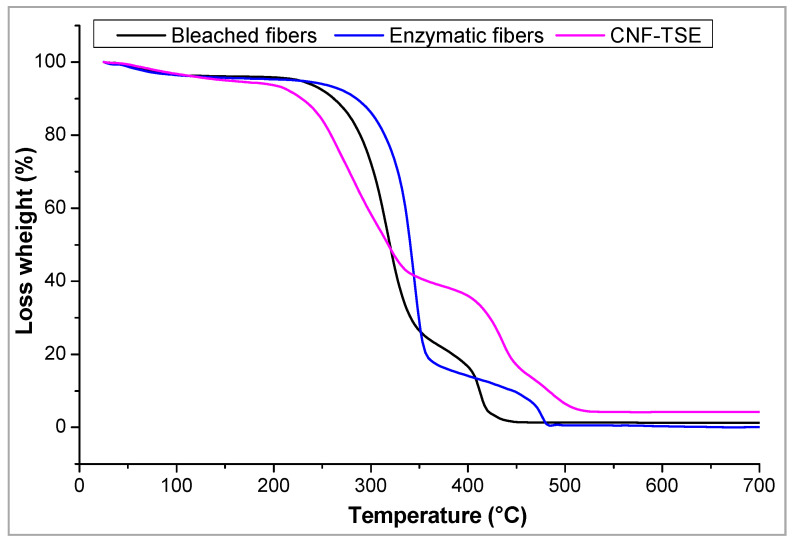
An example of the TGA profiles of the bleached fibers, enzymatic fibers, and CNF-TSE.

**Table 1 molecules-27-06738-t001:** Length, width fine contents, and DP of the obtained fibers from SM.

	Yield (%)	Length (µm)	Width (µm)	Fines Content (%)	DP
Cellulose	58.8	618	25.0	38	618
Enzymatic fiber	51.2	408	18.7	29.7	508
CNF–TSE	49.6	340	19.0	89.2	465
CNF–M	50.2	316	20.0	95.6	275

**Table 2 molecules-27-06738-t002:** Evaluation of the physical properties of the CNF handsheets.

	CNF-M	CNF-TSE
Young Modulus (GPa)	12.31	8.63
Strain at break (%)	0.57	0.69
Tensile strength (kN m^−1^)	3.45	3.57
Thickness for nanopaper (μm)	53.0	64.0
Nanosizer fraction (%)	69.0	52.0
Turbidity (NTU)	494.0	706.0
Quality Index (%)	65.5	63.9

## Data Availability

Not applicable.

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
