# Peer review of "Cellulose Nanofibers from Schinus molle: Preparation and Characterization"

_molecules, 2022, doi:10.3390/molecules27196738_

Round 1

Reviewer 1 Report

Cellulose Nanofibres from Schinus molle: Preparation and Characterization (Molecules -1935874)

The manuscript entitled “Cellulose Nanofibres from Schinus molle: Preparation and Characterization” is an excellent scientific paper comes under green chemistry. The manuscript described accurately the preparation of cellulose nanofiber (CNF) with lignocellulosic biomass of Schinus molle. The CNF was utilized to prepare nano-papers. The preparation of CNF was characterized by various techniques such as X-RD, SEM, optical and atomic force microscopy.

The manuscript can be accepted with minor modifications. Following are the feedback/comments on the manuscript which could be incorporated in the revised manuscript:

1.       2.2. Delignification-Bleaching step: Give the information about starting material and the amount in gram. Furthermore, what was the amount of % moisture released on heating the starting material in Oven. Include the amount of dried material in gram after oven treatment.

2.       Calculate the % yield of cellulose nanofibers in each case and include in Abstract, discussion and conclusion.

3.       Fig.1., Take care in writing the scientific name of the plant. In the figure, include the amount of starting material in g in step 3. In addition, include the amount of the product (g) in last step.

4.       2.5. Nanopapers preparation: Explain the dilution of 15 weight % CNF and write about the solvent and the reason of considering the same solvent.

5.       Include the comparison table for the yield of cellulose Nanofibres (CNF) from Schinus molle with various other reported plants discussed in the manuscript. In addition to this, include the data related with crystallite size of CNF in comparison table.

6.       3.2.2. X-ray diffraction analysis: Include XRD data of “angle at full width at half maximum” for prepared CNF and calculate crystallite size using the relevant published paper (Environmental Monitoring Assessment, year 2021, volume 193: 497) by considering Scherrer’s equation as well as calculator given in data availability section. The same paper can be cited as information provider. The found crystallite size (likely to be calculated) can be compared with other technique crystallite size and can be included in Abstract, discussion and conclusion.

Author Response

Dear colleague,

First of all, I would like to thank you for the time you are spending to review our paper. We tried to take into account all the remarks. The corresponding corrections in the revised manuscript are marked up using “Track Changes” function, and the replies to your queries are listed below.

Cellulose Nanofibres from Schinus molle: Preparation and Characterization (Molecules -1935874)

The manuscript entitled “Cellulose Nanofibres from Schinus molle: Preparation and Characterization” is an excellent scientific paper comes under green chemistry. The manuscript described accurately the preparation of cellulose nanofiber (CNF) with lignocellulosic biomass of Schinus molle. The CNF was utilized to prepare nano-papers. The preparation of CNF was characterized by various techniques such as X-RD, SEM, optical and atomic force microscopy.

Reply: We appreciate the positive feedback from the reviewer. Thank you for taking the time to review our paper and for your valuable comments and compliment.

The manuscript can be accepted with minor modifications. Following are the feedback/comments on the manuscript which could be incorporated in the revised manuscript:

  1. 2.2. Delignification-Bleaching step: Give the information about starting material and the amount in gram. Furthermore, what was the amount of % moisture released on heating the starting material in Oven. Include the amount of dried material in gram after oven treatment.

Reply: Thank you. We have added more information about starting material in section 2.1. More details were added in the revised text.

  1. Calculate the % yield of cellulose nanofibers in each case and include in Abstract, discussion and conclusion.

Reply: Excellent remarks. This information was added in the abstract, results and conclusion section. In addition, this information was reported in table 1 as well as in section 3.1.

  1. Fig.1., Take care in writing the scientific name of the plant. In the figure, include the amount of starting material in g in step 3. In addition, include the amount of the product (g) in last step.

Reply: Thank you. Following your request, we have modified the figure 1 and we have included the used amount in each step.

  1. 2.5. Nanopapers preparation: Explain the dilution of 15 weight % CNF and write about the solvent and the reason of considering the same solvent.

Reply: Thank you. This part was modified as bellow:

“In arrange to get nanopapers, 2 g (of the dry matter CNF) are scattered and diluted with deionized water to 0.5% (w/w) and mixed for 30 minutes under magnetic stirring. At that point, the suspension is filtered using a Rapid Köthen device with the handsheet former equipped with a nylon sieve of 1 μm mesh size, under vacuum. The obtained nanpers are dried between two nylon sieves under vacuum at 45°C for 3 hours. The nanopapers are stored in a conditioned room at 23°C and 50% RH for 48 hours before any characterization.”

  1. Include the comparison table for the yield of cellulose Nanofibres (CNF) from Schinus molle with various other reported plants discussed in the manuscript. In addition to this, include the data related with crystallite size of CNF in comparison table.

Reply: Thank you. We are really sorry, but I confess that we did not understand this remark. In fact, to the best of our knowledge, the yield of CNF is usually equal to 100%, except the loss during the recovery from apparatus (in this study: Twin-screw extruder and Masuko). In addition, It is very hard to make a comparison between the CNF yield and the related cellulose crystallites size. Because, Scherrer’s equation is restricted to samples of high crystallinity and without any broadening of peaks, whereas the CNF described in the literature is very various and which, depending with large parameters such as the source of raw material, CNF preparation, pre-treatment,...So the estimated cellulose crystallites size is not considerable as an intrinsic value and the comparison can not be established.

  1. 3.2.2. X-ray diffraction analysis: Include XRD data of “angle at full width at half maximum” for prepared CNF and calculate crystallite size using the relevant published paper (Environmental Monitoring Assessment, year 2021, volume 193: 497) by considering Scherrer’s equation as well as calculator given in data availability section. The same paper can be cited as information provider. The found crystallite size (likely to be calculated) can be compared with other technique crystallite size and can be included in Abstract, discussion and conclusion.

Thank you. Following your request, the Scherrer’s equation was given and discussed in different part of the revised text.

Scherrer’s Equation was used to estimate the cellulose crystallites' average thickness from their X-ray diffraction patterns [55]:

Where Dhkl is the crystallite dimension normal to the hkl family of lattice planes, K is the correction factor, which is usually 0.94, λ is the radiation wavelength, θ is thediffraction angle, and β1/2 is the peak width at half maximum intensity. Perpendicular to the 0 0 2 planes, the crystal size was determined.

More details were added also in the “results and discussion” section.

We would like to submit our edited revised manuscript. We have considered all the necessary time for the correction of the article by all the authors. We have considered the reviewers’ suggestions and we thank them, because we believe that the quality of our paper was improved. We hope that now our revised paper can meet the required standard of publication in Molecules: Special Issue "Biorefineries".

We thank you in advance and we look forward to hearing from you.

Thank you again for your cooperation.

Best regards,

Younes Moussaoui

Reviewer 2 Report

This article contains many grammatical and scientific errors.

1. Introduction

Line 37…it can be play important in the worldwide ecological … Remark: in this sentence (1) “be” must be deleted, and (2) the word “ecological” should be replaced with “ecology”, i.e. …”it can play important in the worldwide ecology”…

Lines 38-42. Remark: this sentence contains many errors and should be corrected as follows, “Nowadays, the known cellulose product such as nanocellulose has been considered excellent biopolymer that has a large field of applications such as biomedicine [6,19,20], food packaging [21,22], electrochemical performance [23], and also as a giant reinforcement in the matrix [24,25]”. 

Lines 42-44. Remark: this sentence contains many errors and should be corrected as follows, “Two kinds of nanocellulose, cellulose nanocrystals (CNC) and cellulose nanofibrils (CNF), have been extensively researched in terms of isolation, characterization, and application”.

Line 47… are deployed. Remark: this phrase is unnecessary and must be deleted.

 Lines 51-55. Remark: this sentence contains many errors and should be corrected as follows, “The energy-and chemical saves biological pretreatment found in 2007 is now have a great promise for industrial applications due to its environmentally benign conditions and antigenicity of enzymatic response [10,17,30–32]”.

Lines 57-59. Thus numerous research … Remark: (1) the word “Thus” is unnecessary here and must be deleted; start this sentence with “Numerous research…”. (2) correct this sentence as follows, “Numerous research [11,30,32,35,36] concluded that cellulose pretreatment with cellulolytic enzymes cleaves glycosidic linkages, and therefore favors and facilitates the fibrillation of cellulose fibres”.

Lines 59-60. Remark: this sentence contains errors and should be corrected as follows, “The production of CNF involves also mechanical treatment for delaminating the fibres [37–39].

Lines 60-62. Remark: this sentence contains errors and should be corrected as follows, “Currently utilized mechanical equipment can involve high-pressure homogenizers, micro-fluidizers, specialized grinders, and twin-screw extruders [11,40,41]”.

Lines 62-64. Remark: this sentence contains errors and should be corrected as follows, “Along with low density and thermal expansion, CNF have advantages in the development of nanocomposite materials”.

Lines 64-65. Remark: this sentence should be corrected as follows, “NCF can find applications also in water purification and drug delivery [42–48].

Lines 65-66. Remark: this sentence contains many errors and should be corrected as follows, “Diverse plant sources can be used for the production of CNF”.

Line 67-70. Remark: this fragment contains many errors and should be corrected as follows, “The goal of this work is to investigate the viability of generating CNF by treating bleached Schinus molle pulp with enzymes followed by mechanical treatment. The supramolecular structure, morphology, and physicochemical properties of obtained CNF were studied.

2. Materials and Methods

Remark: The authors should shortly describe the little-known plant “Schinus mole” (SM) including the content of cellulose, as well as lignin, hemicelluloses, and extractives. It is also necessary to justify why just this plant sort was chosen for obtaining CNF? What are the specific features and advantages of SM over other types of plant biomass as a raw material for obtaining CNF?

2.2. Remark: This paragraph needs to be corrected, as follows, “The cellulose was isolated from lignocellulosic biomass “Schinus molle” (SM). The lignocellulosic material was collected, washed, and dried at 105 ±5°C to constant weight. Then, the dried material was chopped into uniform pieces about 5 cm in size. Then, the raw material was delignified with 10 wt% NaOH at 160oC for 2 h [49,50]. The resulting fibres were washed thoroughly with distilled water until neutral pH, and bleached with a solution of sodium chlorite at 80±5°C for 3 h. The isolated cellulose was rinsed with distilled water until neutral pH. Finally, the obtained pulp was then stored at 4°C and used to prepare CNF”.

2.3. Remark: The title of 2.3. should be changed, as follows, “Enzymatic pretreatment”

Besides, this paragraph needs to be corrected, as follows, “The enzymatic pretreatment of bleached cellulose was carried out with cellulolytic enzyme “Fiber Care R” having a dose of 300 ECU per 1 g of cellulose. The 2 wt. % cellulose dispersion in acetate buffer (pH=5) was treated with the enzyme solution with stirring at a speed of 200 rpm at 50°C for 2 hours. After that, the dispersion was heated to 80°C for 15 min, and then cooled to 25°C. Finally, the treated cellulose was washed thoroughly with distilled water until neutral pH and isolated using a 1 µm nylon mesh filter, and stored at 4° C for further characterization and preparation of CNF”.

2.5. Line 111-120. Remark: Correct this fragment, as follows, “The 1 wt% dispersion of CNF was stirred with an ultra-turrax for 1 min and then isolated using a 1 µm nylon mesh filter. To prepare nanopaper, a Rapid Köthen apparatus was used according to the IS0 5269-2 standard method. Wet sheets of nanopaper were placed between two protective sheets and dried at 85°C for 20 min. Prior to characterization, the sheets of dried nanopaper were conditioned at 23° C and 50% relative humidity for 48 h.

Figure 1 shows the scheme for the preparation of CNF.

2.6. Remark: In this section, the word “fibre” needs plural form, i.e., “fibres”.

Further, the sentence “In the bellow, we decided to describe in briefly” is unnecessary and must be deleted. The next sentence should be corrected, as follows, “A light microscope Carl Zeiss Axio Imager M1m was used”.

Line 128. Remark: Replace “cellulose suspension” with “cellulose dispersion”.

Line 130. Remark: Replace “a pulp solution” with “a pulp dispersion”.

Line 132. Remark: Correct this sentence, as follows, “The morphological structure of CNF was studied using the Transmission”…

Line 134. Remark: Replace “diluted suspension” with “diluted dispersion”.

Lines 137-138. Remark: Correct this sentence, as follows, “The AFM analysis was performed using a tapping mode of OTESPA cantilever”.

Lines 138-139.  Remark: Correct this sentence, as follows, “A CNF dispersion (10-3 weight percent) was spread out on a mica disk and allowed to air dry”.

Line 139. Remark: Replace “CNF suspension” with “CNF dispersion”.

Line 144. ….of 100 of CNF. Remark: Delete repeated “of” and write …” of 100 CNF”.

Line 150. …spectra was.. Remark: Replace “was” with “were”.

Lines 151-152. Remark: Delete the sentence “The cellulose's crystallinity index (CI) can be estimated”, since it is unnecessary. Correct the next sentence, as follows, “The crystallinity index (CI) was determined according to the Segal et al. method [54].

Line 157. Remark: Delete the sentence end, …“using the capillary viscometer method”, since it is unnecessary.

Line 161. Remark: Replace “suspension” with “dispersion”.

Line 162. Remark: Correct this sentence, as follows, “The nanoscale fraction was estimated according to method of Naderi et al [55]”.

Line 163. Remark: Correct this sentence, as follows, “Three tests were run, and the average value was then found”.

Line 164. Remark: Replace “suspensions” with “dispersions”.

Line 177.  Remark: The authors should note that eq. (4) can only be used for round paper samples with a diameter of d. The general equation to calculate the grammage of the paper sheet is the following: G= m/S, where m is the weight of the paper sheet (g), and S is the surface area of the paper sheet expressed in m2 (see TAPPI Standard T 410).

Line 178. Remark: Correct this sentence, as follows, “In addition, the thickness of nanopaper sheets was measured”

Line 185. Remark: Correct this sentence, as follows, “The mechanical properties of nanopapers were tested”.

Line 192. Remark: What denotes symbol l?

3. Results and discussion

Line 194. Remark: The verb “were” was missed, and this sentence should be corrected, as follows, “Cellulose fibres with high purity and crystallinity were extracted from Schinus molle using alkaline and bleaching treatments”.

Line 195. Remark: Correct also this sentence: “From these fibres…”

Line 198. Remark: Correct this sentence, as follows, “It can improve”…

Line 199. Remark: Correct this sentence, as follows, “The enzymatic pretreatment of cellulose  can reduce the degree of polymerization and…”

Table 1. Remark: The authors should explain why the width of CNF is 19-20 µm instead of 19-20 nm? If the width of CNF actually is 19-20 µm, then they are microfibers and not nanofibers. In addition, these results for CNF contradict the data of morphological analysis (see section 3.2.1) according to which the width of CNF is 20-30 nm. However, if Table 1 contains results for delignified cellulose and enzymatically treated cellulose, the results for CNF-TSE and CNF-M must be excluded

Lines 226, 237. For fibres treated with 15 wt% NaOH… use NaOH 15 wt%... Remark: If, in fact, the authors treated the fibers with 15 wt.% NaOH, then why was this treatment not described in section 2 or in Fig. 1? This omission must be corrected. Authors should describe the conditions of this treatment including temperature and duration.

Line 253. Remark: the word “degree” is missing, namely authors should write …”polymerization degree”…

Line 267. Remark: the word “DP” is missing, namely authors should write “We saw a 59% reduction in DP for the enzymatic pulp”…

Section 3.2. Characterization of CNF

Line 279. The extent of enzymatic degradation of the fibres… Remark: Since sections 3.2 and 3.2.1 describe CNF, the discussion of enzymatic degradation of the fibres should be excluded from this section. Instead, the authors should write, “The morphology of CNF was studied by different methods, such as optical microscopy, fiber analyzer, and AFM”…

Line 281. Remark: The number of this Figure must be Fig. 3 and not Fig. 2.

Lines 286-287. Remark: This sentence should be corrected, as follows, “The uniform size distribution was observed for CNF.

Line 296. Remark: This sentence should be corrected, as follows, “A small nano fraction of the CNF materials was observed (Fig. 3 a, b)”…

Lines 297-298. The enzymatic treatment causes a significant reduction in the fibre length, whereas the fibre diameter was not much affected since some fibres were not fibrillated during the process. Remark: Since section 3.2.1 describes CNF, the discussion of enzymatic degradation of the fibres should be excluded. Therefore, this sentence must be deleted.

Lines 301-309. Remark: Since section 3.2.1 describes CNF, the discussion on enzymatic hydrolysis of the fibres should be excluded from this section. Thus, the whole fragment between lines 301 and 309 must be deleted or moved in section 3.1 where enzymatically treated fibres were discussed.

Line 326. Remark: The term “crystalline shape” is not suitable here and should be replaced with “crystalline allomorph”.

Line 389.  Remark: Replace “CNF suspensions” with “CNF dispersions”.

Decision:  The English of the authors is poor. In addition, the article contains many scientific errors. Therefore, this article cannot be recommended for publication in its present form and needs major revision.

Author Response

Dear colleague,

First of all, I would like to thank you for taking the time to review our paper and for your valuable comments and compliment. The corresponding corrections in the revised manuscript are marked up using “Track Changes” function, and the replies to your queries are listed below.

This article contains many grammatical and scientific errors.

Reply: Thank you. We have revised the whole manuscript carefully and tried to avoid any grammar or syntax error. In addition, we have asked several colleagues who are ESP teacher in our faculty. We believe that the language is now acceptable.

  1. Introduction
  • Line 37…it can be play important in the worldwide ecological … Remark: in this sentence (1) “be” must be deleted, and (2) the word “ecological” should be replaced with “ecology”, i.e. …”it can play important in the worldwide ecology”…

Reply: Thank you. This was corrected.

  • Lines 38-42. Remark: this sentence contains many errors and should be corrected as follows, “Nowadays, the known cellulose product such as nanocellulose has been considered excellent biopolymer that has a large field of applications such as biomedicine [6,19,20], food packaging [21,22], electrochemical performance [23], and also as a giant reinforcement in the matrix [24,25]”.

Reply: Thank you. This was modified.

  • Lines 42-44. Remark: this sentence contains many errors and should be corrected as follows, “Two kinds of nanocellulose, cellulose nanocrystals (CNC) and cellulose nanofibrils (CNF), have been extensively researched in terms of isolation, characterization, and application”.

Reply: Thank you. This was done.

  • Line 47… are deployed. Remark: this phrase is unnecessary and must be deleted.

Reply: Thank you. This was revised.

  • Lines 51-55. Remark: this sentence contains many errors and should be corrected as follows, “The energy-and chemical saves biological pretreatment found in 2007 is now have a great promise for industrial applications due to its environmentally benign conditions and antigenicity of enzymatic response [10,17,30–32]”.

Reply: Thank you. This was corrected.

  • Lines 57-59. Thus numerous research … Remark: (1) the word “Thus” is unnecessary here and must be deleted; start this sentence with “Numerous research…”. (2) correct this sentence as follows, “Numerous research [11,30,32,35,36] concluded that cellulose pretreatment with cellulolytic enzymes cleaves glycosidic linkages, and therefore favors and facilitates the fibrillation of cellulose fibres”.

Reply: Thank you. This was done.

  • Lines 59-60. Remark: this sentence contains errors and should be corrected as follows, “The production of CNF involves also mechanical treatment for delaminating the fibres [37–39].

Reply: Thank you. This was modified.

  • Lines 60-62. Remark: this sentence contains errors and should be corrected as follows, “Currently utilized mechanical equipment can involve high-pressure homogenizers, micro-fluidizers, specialized grinders, and twin-screw extruders [11,40,41]”.

Reply: Thank you. This was done.

  • Lines 62-64. Remark: this sentence contains errors and should be corrected as follows, “Along with low density and thermal expansion, CNF have advantages in the development of nanocomposite materials”.

Reply: Thank you. This was corrected.

  • Lines 64-65. Remark: this sentence should be corrected as follows, “NCF can find applications also in water purification and drug delivery [42–48]”.

Reply: Thank you. This was done.

  • Lines 65-66. Remark: this sentence contains many errors and should be corrected as follows, “Diverse plant sources can be used for the production of CNF”.

Reply: Thank you. This was corrected.

  • Line 67-70. Remark: this fragment contains many errors and should be corrected as follows, “The goal of this work is to investigate the viability of generating CNF by treating bleached Schinus molle pulp with enzymes followed by mechanical treatment. The supramolecular structure, morphology, and physicochemical properties of obtained CNF were studied”.

Reply: Thank you for your comments. We have revised the introduction and all remarks were considered in the revised manuscript.

  1. Materials and Methods
  • Remark: The authors should shortly describe the little-known plant “Schinus mole” (SM) including the content of cellulose, as well as lignin, hemicelluloses, and extractives. It is also necessary to justify why just this plant sort was chosen for obtaining CNF? What are the specific features and advantages of SM over other types of plant biomass as a raw material for obtaining CNF?

Reply: Thank you. Fowling your suggestion, more information data about the raw material and the advantage of use to produce CNF were added in the revised text (please see the material in section 2.1).

  • 2.2. Remark: This paragraph needs to be corrected, as follows, “The cellulose was isolated from lignocellulosic biomass “Schinus molle” (SM). The lignocellulosic material was collected, washed, and dried at 105 ±5°C to constant weight. Then, the dried material was chopped into uniform pieces about 5 cm in size. Then, the raw material was delignified with 10 wt% NaOH at 160oC for 2 h [49,50]. The resulting fibres were washed thoroughly with distilled water until neutral pH, and bleached with a solution of sodium chlorite at 80±5°C for 3 h. The isolated cellulose was rinsed with distilled water until neutral pH. Finally, the obtained pulp was then stored at 4°C and used to prepare CNF”.

Reply: Very sorry for these multiples’ mistakes. This was corrected. Thank you.

  • 2.3. Remark: The title of 2.3. should be changed, as follows, “Enzymatic pretreatment”

Reply: Thank you. This was corrected.

  • Besides, this paragraph needs to be corrected, as follows, “The enzymatic pretreatment of bleached cellulose was carried out with cellulolytic enzyme “Fiber Care R” having a dose of 300 ECU per 1 g of cellulose. The 2 wt. % cellulose dispersion in acetate buffer (pH=5) was treated with the enzyme solution with stirring at a speed of 200 rpm at 50°C for 2 hours. After that, the dispersion was heated to 80°C for 15 min, and then cooled to 25°C. Finally, the treated cellulose was washed thoroughly with distilled water until neutral pH and isolated using a 1 µm nylon mesh filter, and stored at 4° C for further characterization and preparation of CNF”.

Reply: Thank you. This part was modified.

  • 2.5. Line 111-120. Remark: Correct this fragment, as follows, “The 1 wt% dispersion of CNF was stirred with an ultra-turrax for 1 min and then isolated using a 1 µm nylon mesh filter. To prepare nanopaper, a Rapid Köthen apparatus was used according to the IS0 5269-2 standard method. Wet sheets of nanopaper were placed between two protective sheets and dried at 85°C for 20 min. Prior to characterization, the sheets of dried nanopaper were conditioned at 23° C and 50% relative humidity for 48 h.

Figure 1 shows the scheme for the preparation of CNF.

Reply: Thank you. This part was modified.

  • 2.6. Remark: In this section, the word “fibre” needs plural form, i.e., “fibres”.

Reply: Thank you. This was done.

  • Further, the sentence “In the bellow, we decided to describe in briefly” is unnecessary and must be deleted. The next sentence should be corrected, as follows, “A light microscope Carl Zeiss Axio Imager M1m was used”.

Reply: Thank you. This was done.

  • Line 128. Remark: Replace “cellulose suspension” with “cellulose dispersion”.

Reply: Thank you. This was modified.

  • Line 130. Remark: Replace “a pulp solution” with “a pulp dispersion”.

Reply: Thank you. This was done.

  • Line 132. Remark: Correct this sentence, as follows, “The morphological structure of CNF was studied using the Transmission”…

Reply: Thank you. This was corrected.

  • Line 134. Remark: Replace “diluted suspension” with “diluted dispersion”.

Reply: Thank you. This was done.

  • Lines 137-138. Remark: Correct this sentence, as follows, “The AFM analysis was performed using a tapping mode of OTESPA cantilever”.

Reply: Thank you. This was corrected.

  • Lines 138-139. Remark: Correct this sentence, as follows, “A CNF dispersion (10-3 weight percent) was spread out on a mica disk and allowed to air dry”.

Reply: Thank you. This was done.

  • Line 139. Remark: Replace “CNF suspension” with “CNF dispersion”.

Reply: Thank you. This was done.

  • Line 144. ….of 100 of CNF. Remark: Delete repeated “of” and write …” of 100 CNF”.

Reply: Thank you. This was revised.

  • Line 150. …spectra was.. Remark: Replace “was” with “were”.

Reply: Thank you. This was corrected.

  • Lines 151-152. Remark: Delete the sentence “The cellulose's crystallinity index (CI) can be estimated”, since it is unnecessary. Correct the next sentence, as follows, “The crystallinity index (CI) was determined according to the Segal et al. method [54].

Reply: Thank you. This was corrected.

  • Line 157. Remark: Delete the sentence end, …“using the capillary viscometer method”, since it is unnecessary.

Reply: Thank you. This was done.

  • Line 161. Remark: Replace “suspension” with “dispersion”.

Reply: Thank you. This was corrected.

  • Line 162. Remark: Correct this sentence, as follows, “The nanoscale fraction was estimated according to method of Naderi et al [55]”.

Reply: Thank you. This was modified.

  • Line 163. Remark: Correct this sentence, as follows, “Three tests were run, and the average value was then found”.

Reply: Thank you. This was corrected.

  • Line 164. Remark: Replace “suspensions” with “dispersions”.

Reply: Thank you. This was done.

  • Line 177. Remark: The authors should note that eq. (4) can only be used for round paper samples with a diameter of d. The general equation to calculate the grammage of the paper sheet is the following: G= m/S, where m is the weight of the paper sheet (g), and S is the surface area of the paper sheet expressed in m2 (see TAPPI Standard T 410).

Reply: Thank you. This was done.

  • Line 178. Remark: Correct this sentence, as follows, “In addition, the thickness of nanopaper sheets was measured”

Reply: Thank you. This was corrected and revised.

  • Line 185. Remark: Correct this sentence, as follows, “The mechanical properties of nanopapers were tested”.

Reply: Thank you. This was modified.

  • Line 192. Remark: What denotes symbol l?

Reply: Thank you. Thank you. We have deleted equation (5) because breaking length (LR) was not discussed in the manuscript.

  1. Results and discussion
  • Line 194. Remark: The verb “were” was missed, and this sentence should be corrected, as follows, “Cellulose fibres with high purity and crystallinity were extracted from Schinus molle using alkaline and bleaching treatments”.

Reply: Thank you. This was corrected.

  • Line 195. Remark: Correct also this sentence: “From these fibres…”

Reply: Thank you. This was modified.

  • Line 198. Remark: Correct this sentence, as follows, “It can improve”…

Reply: Thank you. This was done.

  • Line 199. Remark: Correct this sentence, as follows, “The enzymatic pretreatment of cellulose can reduce the degree of polymerization and…”

Reply: Thank you. This was revised and corrected.

  • Table 1. Remark: The authors should explain why the width of CNF is 19-20 µm instead of 19-20 nm? If the width of CNF actually is 19-20 µm, then they are microfibers and not nanofibers. In addition, these results for CNF contradict the data of morphological analysis (see section 3.2.1) according to which the width of CNF is 20-30 nm. However, if Table 1 contains results for delignified cellulose and enzymatically treated cellulose, the results for CNF-TSE and CNF-M must be excluded.

Reply: Excellent remarks. Very sorry for these mistakes. More details were added in revised text.

  • Lines 226, 237. For fibres treated with 15 wt% NaOH… use NaOH 15 wt%... Remark: If, in fact, the authors treated the fibers with 15 wt.% NaOH, then why was this treatment not described in section 2 or in Fig. 1? This omission must be corrected. Authors should describe the conditions of this treatment including temperature and duration.

Reply: Thank you for pointing this out. Sorry for this mistake, treatment was done using 10 wt% NaOH. This was corrected in the revised manuscript.

  • Line 253. Remark: the word “degree” is missing, namely authors should write …”polymerization degree”…

Reply: Thank you. This was done.

  • Line 267. Remark: the word “DP” is missing, namely authors should write “We saw a 59% reduction in DP for the enzymatic pulp”…

Reply: Thank you. This was corrected.

Section 3.2. Characterization of CNF

  • Line 279. The extent of enzymatic degradation of the fibres… Remark: Since sections 3.2 and 3.2.1 describe CNF, the discussion of enzymatic degradation of the fibres should be excluded from this section. Instead, the authors should write, “The morphology of CNF was studied by different methods, such as optical microscopy, fiber analyzer, and AFM”…

Reply: Thank you. This was revised.

  • Line 281. Remark: The number of this Figure must be Fig. 3 and not Fig. 2.

Reply: Thank you. This was corrected.

  • Lines 286-287. Remark: This sentence should be corrected, as follows, “The uniform size distribution was observed for CNF”.

Reply: Thank you. This was done.

  • Line 296. Remark: This sentence should be corrected, as follows, “A small nano fraction of the CNF materials was observed (Fig. 3 a, b)”…

Reply: Thank you. This was modified.

  • Lines 297-298. The enzymatic treatment causes a significant reduction in the fibre length, whereas the fibre diameter was not much affected since some fibres were not fibrillated during the process. Remark: Since section 3.2.1 describes CNF, the discussion of enzymatic degradation of the fibres should be excluded. Therefore, this sentence must be deleted.

Reply: Thank you. This was revised.

  • Lines 301-309. Remark: Since section 3.2.1 describes CNF, the discussion on enzymatic hydrolysis of the fibres should be excluded from this section. Thus, the whole fragment between lines 301 and 309 must be deleted or moved in section 3.1 where enzymatically treated fibres were discussed.

Reply: Thank you. This was revised and corrected.

  • Line 326. Remark: The term “crystalline shape” is not suitable here and should be replaced with “crystalline allomorph”.

Reply: Thank you. This was revised and corrected.

  • Line 389. Remark: Replace “CNF suspensions” with “CNF dispersions”.

Reply: Thank you. This was revised and corrected.

We would like to submit our edited revised manuscript. We have considered all the necessary time for the correction of the article by all the authors. We have considered the reviewers’ suggestions and we thank them, because we believe that the quality of our paper was improved. We hope that now our revised paper can meet the required standard of publication in Molecules: Special Issue "Biorefineries".

We thank you in advance and we look forward to hearing from you.

Thank you again for your cooperation.

Best regards,

Younes Moussaoui

Round 2

Reviewer 2 Report

Review

The revised article can be published with minor corrections.

I recommend using the terms “fiber” and “fibers” instead of “fibre” and “fibres”

Regarding the chemical composition of Schinus mole (section 2.1). Remark:  I am surprised that the amount of extractives dissolved in cold water (38.2%) is greater than in hot water (27.0%), and 1% NaOH (27.0%). Is this a mistake result that needs to be corrected?

Regarding the formation of nanopaper (section 2.5). Line 202. Water has been chosen as a solvent; Remark:  Water is not solvent for cellulose; therefore, you need to replace the word “solvent” with “diluent”.

Author Response

Dear colleague,

We would like to thank you for the time you have spent to review our paper and for your constructive comments and suggestions. The corresponding corrections in the revised manuscript are marked up using “Track Changes” function, and the replies to your queries are listed below.

The revised article can be published with minor corrections.

Reply: Thank you. We appreciate your positive feedback.

I recommend using the terms “fiber” and “fibers” instead of “fibre” and “fibres”

Reply: Thank you for your comment. We have revised the whole manuscript and we have modified “fibre” and “fibres” by “fiber” and “fibers”.

Regarding the chemical composition of Schinus mole (section 2.1). Remark:  I am surprised that the amount of extractives dissolved in cold water (38.2%) is greater than in hot water (27.0%), and 1% NaOH (27.0%). Is this a mistake result that needs to be corrected?

Reply: Thank you. Very sorry for this mistake. This was corrected as bellow:

“The amount of extractive solubility in hot water, in 1% NaOH, and in ethanol–toluene was 38.2, 27.0, and 12.3 %, respectively.”

Regarding the formation of nanopaper (section 2.5). Line 202. Water has been chosen as a solvent; Remark:  Water is not solvent for cellulose; therefore, you need to replace the word “solvent” with “diluent”.

Reply: Thank you. This was corrected.

We would like to submit our edited revised manuscript. We hope that now our revised paper can meet the required standard of publication in Molecules: Special Issue "Biorefineries".

We thank you in advance and we look forward to hearing from you.

Thank you again for your cooperation.

Best regards,

Younes Moussaoui

Round 3

Reviewer 2 Report

The additionally revised article can be recommended for publication